# From Novice to Master(s) Level Athlete: A Longitudinal Analysis of Psychological Changes in a Marathon Runner Completing 119 Marathons

**DOI:** 10.3390/bs15070893

**Published:** 2025-06-30

**Authors:** Xiuxia Liu, Lisheng Huang, Shunying Lin

**Affiliations:** Department of Physical Education, Xiamen University, Xiamen 361005, China; liuxiuxia@xmu.edu.cn (X.L.); huanglisheng@xmu.edu.cn (L.H.)

**Keywords:** marathon running, psychological evolution, longitudinal case study, identity transition, socio-cultural context

## Abstract

Long-term participation in marathon running involves complex psychological processes, yet existing research predominantly focuses on static, single-time-point analyses. This study addresses the gap by longitudinally examining the psychological evolution of an elite Chinese marathon runner (119 marathons completed) to uncover dynamic shifts from novice to master(s) level athlete stages. A longitudinal single-case study was conducted using inductive thematic analysis. Data included in-depth interviews, observational records, and archival materials spanning three life stages (youth, middle age, maturity). Five experts validated the credibility and validity of the findings. The results show that the runner’s psychological trajectory followed a three-phase model: competitive drive (youth: external achievement motivation), reflective transformation (middle age: health prioritization and identity reconfiguration), and value reconstruction (maturity: legacy mission and lifelong running). These stages were shaped by the interplay of achievement motivation, social roles, and physiological changes. Notably, the transition mirrored China’s marathon culture shift from elitism to mass participation. This study proposes a novel “motivation-physicality-society” interaction model, challenging static theories of sports psychology. It highlights how long-term runners dynamically balance extrinsic and intrinsic motivations while embedding personal growth within socio-cultural transformations. The findings offer theoretical foundations for optimizing psychological support systems and promoting sustainable marathon engagement.

## 1. Introduction

In recent years, the popularization of marathon running has become increasingly evident. This sport is a physical challenge. With the growing number of participants, it has become significantly important to study the psychological dynamics of marathon runners. Previous research has mainly focused on exploring the motivations of marathon participants through questionnaires and interviews ([2]; [11]; [28]).

However, most existing studies focus on a single life stage, resulting in fragmented research on psychological states. Few studies have examined the dynamic changes across the entire life span, primarily due to the lack of longitudinal tracking of long-term participants. Current theories fail to fully explain the stage-specific characteristics and turning mechanisms of psychological changes in marathon runners. This gap limits our understanding of the psychological dynamics underlying sustained participation in marathon running and hampers the promotion of this sport.

The present study aims to fill this gap by conducting a longitudinal analysis of the psychological evolution of long-term marathon runners, focusing on the transition from novice to master(s) level athlete stages. Additionally, systematic reviews have emphasized the need for in-depth psychological studies of remarkable athletes ([20]). This study will focus on three main aspects: (1) the psychological characteristics and their evolution at different stages of marathon running; (2) the impact of these psychological changes on sustained participation in marathon running; and (3) the manifestation of marathon spirit in the development of marathon culture in China. This study will be the first to construct a longitudinal psychological tracking model based on 119 marathons, providing a new perspective for understanding the psychological changes in the number of participants. The findings will offer theoretical foundations for optimizing training systems for amateur runners and for health behavior interventions, thereby promoting the sustainable development of marathon running.

## 2. Literature Review

### 2.1. Motivations of Marathon Runners

Previous research has mainly focused on exploring the motivations of marathon participants through questionnaires and interviews ([2]; [11]). These studies have identified motivational factors such as health ([14]), gender, and age differences among marathon and ultramarathon runners ([19]) and psychological resilience ([20]). Other factors include mental toughness, grit, motivation, general self-efficacy, and personality traits ([13]), as well as motivation, personality, and trait self-handicapping ([26]). Additionally, research has examined the sense of identity, group belonging ([21]), and social support ([6]). Notably, female runners tend to be more motivated by personal goal achievement and external recognition, while male runners are more driven by personal satisfaction and well-being from ultrarunning ([2]). The experience of marathon runners has also been characterized by five features: preparation and strategy, management, discovery, personal achievement, and community involvement ([22]).

### 2.2. Psychological Characteristics and Identity

Beyond motivations, the psychological characteristics and identity of marathon runners have been extensively studied. [21] ([21]) demonstrated that participation in marathons fosters a unique social identity and sense of belonging among runners. This identity not only enhances psychological resilience but also promotes sustained engagement in the sport. [6] ([6]) further explored the role of social support in shaping runners’ psychological states, revealing that support from family, friends, and fellow runners significantly improves mental health and motivation. The dynamic nature of identity has also been highlighted in the context of marathon running. According to [23]’s ([23]) identity control theory, identity is a dynamic construct that adjusts in response to social feedback and role transitions. This theory provides a valuable lens for understanding how runners’ identities evolve across different stages of their running careers.

### 2.3. Dynamic Nature of Psychological Evolution

Despite existing insights into the psychological characteristics of marathon runners, most studies have focused on specific stages or groups, lacking longitudinal tracking of psychological changes in runners who participate in multiple marathons over an extended period. Recent reviews have emphasized the need for longitudinal research to capture the dynamic nature of runners’ psychological evolution. [20] ([20]) highlighted the complexity of long-term engagement in marathons and noted that current theories inadequately explain the stage-specific features and turning points in runners’ psychological trajectories. Longitudinal studies, such as those by [16] ([16]), have shown that life cycle events significantly influence the dynamics of motivation. These findings underscore the importance of longitudinal research to fill the gap in understanding the psychological evolution of marathon runners.

### 2.4. Cultural and Societal Influences

The popularization of marathon running has transformed it into a significant sociocultural phenomenon. The shift from elitism to mass participation has had profound effects on runners’ psychological profiles and motivations. [12]’s ([12]) structuration theory, which emphasizes the dynamic interplay between individual practices and social structures, provides a framework for understanding these sociocultural influences. Additionally, [1]’s ([1]) Selective Optimization with Compensation (SOC) theory offers insights into how runners adapt to age-related physiological changes by reallocating resources and adjusting goals. This theory is particularly relevant for understanding the psychological adaptations of marathon runners across different life stages. In light of the preceding discussion, we posit the following hypothesis:

**Hypothesis** **1.**
*The interplay among motivation, the physical self, and societal factors offers a more comprehensive and dynamic explanation of the psychological evolution of marathon runners. This framework elucidates the reciprocal relationships between motivation, physical changes, and social influences, as well as their collective impact on runners’ psychological states.*


### 2.5. Research Gaps and Future Directions

Despite significant progress in understanding the motivations and psychological characteristics of marathon runners, several research gaps remain. First, the majority of existing studies employ cross-sectional designs, limiting the understanding of the dynamic evolution of runners’ psychological states. Second, current theories inadequately explain the stage-specific features and turning mechanisms in runners’ psychological trajectories. Future research should address these gaps by adopting multi-case designs. Longitudinal cohort studies are particularly needed to provide deeper insights into the psychological evolution of marathon runners and to inform targeted psychological support and training interventions.

In conclusion, the psychological characteristics and evolution of marathon runners represent a complex and multidimensional research domain. Existing studies have illuminated the roles of achievement motivation, psychological resilience, identity, and social support in shaping runners’ experiences. However, these studies are often limited to specific stages and lack longitudinal tracking. This research should explore the dynamic psychological evolution of marathon runners across their entire life span and elucidate stage-specific features and turning points to provide a comprehensive theoretical framework for understanding the psychological trajectories of marathon runners.

Based on the literature review above, the marathon runner who completed 119 marathons experienced psychological changes from the novice stage through the intermediate stage to the master(s) level. The following hypotheses are proposed:

**Hypothesis** **2.**
*There are differences in the psychological characteristics of marathon runners across different stages from novice to master(s) level.*


**Hypothesis** **3.**
*Novice marathon runners are primarily driven by external achievement motivation, such as the pursuit of competitive performance.*


**Hypothesis** **4.**
*Intermediate-level marathon runners shift their motivation toward prioritizing health and reconstructing their identity.*


**Hypothesis** **5.**
*At the master(s) level, marathon runners are mainly motivated by the mission of perpetuating the marathon spirit and lifelong running.*


## 3. Methods

### 3.1. Participants

This study utilized a longitudinal case study approach, selecting an elite Chinese marathon runner as the subject. The subject, hereafter referred to as Mr. A, is a 57-year-old associate professor at the Physical Education Department of Xiamen University. Mr. A had not received formal coaching for marathon running but had been trained in sprinting. This background in sprint training can be considered as a form of relevant instruction. His initial motivation was a combination of external and internal factors. Externally, his coach’s expectations played a role, while internally, he aspired to achieve good athletic performance results. He completed his first marathon at the age of 35 and has since finished 119 full marathons globally, including those in Boston, USA; Vancouver, Canada; Gold Coast, Australia; and Osaka, Japan. Mr. A also serves as a course instructor for marathon programs, an ambassador for the sports brand X-tep, and a certified “lifelong runner” by World Athletics. As a prominent advocate for mass marathon participation in China, his extensive experience and significant influence render him an ideal candidate for case study research.

### 3.2. Procedure

As a colleague of Mr. A, the researchers conducted multiple in-depth interviews with him, covering his running experiences, psychological states, motivational changes, and insights into marathon running across different life stages. The semi-structured interviews were designed to thoroughly explore his inner experiences and behavioral motivations. Two researchers participated in each interview session, with one asking questions and the other taking detailed notes to ensure the completeness and accuracy of the content. With the consent of the interviewee, the interviews were recorded and transcribed verbatim afterward to maintain the originality and authenticity of the data. In addition, interviews lasted approximately one hour. The interview questions primarily included: “What were your initial motivations for running marathons?” “How have these motivations changed over time? Can you delineate these changes by stages?” “What has sustained your commitment to complete 119 marathons?”

At the same time, the researchers collected Mr. A’s academic papers, publications, and media reports from various stages of his life. These materials provided rich contextual information and supplementary evidence, aiding in a comprehensive understanding of the trajectory of his psychological evolution. Through systematic organization and analysis of these documents, the researchers identified key milestones and psychological turning points in his running career.

Additionally, adopting [24]’s ([24]) life-span theory, this study divided the psychological evolution of Mr. A into three phases: novice (1980–2003), intermediate (2003–2010), and master(s) level athlete (2010–present). The study analyzed the psychological characteristics of each phase and explored the psychological changes from novice to veteran status. This method enabled an in-depth investigation of the psychological dynamics underlying long-term marathon participation, filling the gap in existing research regarding the longitudinal tracking of marathon runners’ psychological changes.

### 3.3. Data Analysis

In the present study, data were analyzed using inductive thematic analysis ([3]; [4]). Initially, researchers thoroughly reviewed the interview transcripts, the literature, and observational notes to familiarize themselves with the data and identify preliminary themes and patterns. Subsequently, the data were coded into distinct themes, each named and defined based on key concepts and phenomena within the dataset. These coded data were then integrated and refined to generate representative and explanatory analytical themes.

To ensure the credibility and validity of the findings, the first author convened a five-member analysis team, comprising four additional experts, to undertake the data coding process. Each member of this team is a doctoral candidate with a background in conducting interview-based research and a publication record in this domain and also with extensive experience and a strong theoretical foundation in marathon studies. Their collective expertise ensures a robust and reliable coding process for this research endeavor. Following collective discussion and review, the credibility and validity of the findings were confirmed, ensuring the scientific rigor and reliability of the study conclusions.

In addition, to address potential biases arising from the first author’s professional relationship with interviewee A, we implemented several measures to ensure the objectivity and impartiality of the interview. Firstly, we adopted anonymous recording during the interview process. Additionally, we invited a researcher who has no direct connection with either the first author or the interviewee to observe and document the interview. This arrangement helps to maintain the integrity of the interview data.

In the data analysis phase, we employed multiple methods to cross-validate the data. For instance, we analyzed and compared publicly available television interviews and newspaper reports about Mr. A to minimize interpretative biases that might stem from personal relationships. Moreover, four experts were invited to collaborate with the first author in the data analysis process, further ensuring the objectivity of the research findings.

It should be noted that this study obtained ethical approval from our institution, which deemed there were no ethical concerns. Mr. A, a veteran teacher nearing retirement, has a profound passion for marathons. He expressed his willingness to share all relevant information openly and honestly, without any concealment or sensitivity issues. These factors collectively alleviate potential reader concerns regarding ethical issues.

In the coding process, open coding was initially conducted to extract sentences from Mr. A’s marathon running experiences that were linked to his psychological feelings, such as his statement in this study’s interview: “Now, health comes first. I prioritize body maintenance.” These sentences were then identified as content for further analysis and comparison. Subsequently, axial coding was performed to refine the extracted sentences from the open coding, resulting in more concise concepts like “Health priority and sustainable running.” Finally, selective coding was applied to compare and distill the axial coding outcomes into more abstract meaning units, such as “Health-focused transformation.”

Through this methodological approach, the study systematically analyzed the psychological evolution of Mr A from a novice to a master(s) level athlete, revealing the psychological patterns and underlying mechanisms associated with long-term marathon participation.

## 4. Results

### 4.1. Novice Period (1980–2003): Competitive Mindset and Self-Breakthrough

Table 1 presents the results of the psychological characteristics analysis of Mr. A during the novice period.

As shown in Table 1, the results indicate that during the novice period, Mr. A exhibited a strong competitive mindset and motivation for self-breakthrough in his engagement with marathon running. This aligns with achievement motivation theory ([17]), which suggests that novice runners are primarily driven by external achievement motives, striving for performance outcomes and recognition.

The competitive mindset was characterized by his self-demands as a professional athlete and his tenacious spirit. Mr. A, hailing from a family with a strong sports background—his father being a renowned track and field athlete in Fujian Province—developed a keen competitive consciousness from a young age. During his middle school years, he dominated school track meets, winning the 400 m, 800 m, and 1500 m events, and continued to achieve notable successes in provincial and national competitions (e.g., 800 m champion at the Fujian Provincial Games, national youth runner-up). His psychological profile during this stage was centered on the pursuit of speed and performance breakthroughs, reflecting the self-expectations of a professional athlete and embodying the spirit of “faster, higher, stronger” in competitive sports. As he mentioned in the interview, “My first marathon was driven by my lifelong passion for running, my background as a professional runner, and my academic focus on running. Despite being a specialist in the 800m track event and being 35 years old, I felt compelled to participate actively in the inaugural Xiamen Marathon.”

Moreover, Mr. A derived a strong sense of achievement from overcoming challenges, continually reinforcing his tenacious spirit. He emphasized in the interview, “Running a marathon is not just about running; it’s about cultivating perseverance and a strong will. In middle school, I was so determined to win that I would have pushed myself to the limit.”

Self-breakthrough was manifested in three aspects: First, he took pride in completing arduous tasks, as he stated, “Each full marathon is tough, but finishing it brings immense satisfaction and a sense of accomplishment.” Second, he adhered to a self-reliant approach, noting, “I may encounter minor injuries and sometimes get sick, but my name is hardworking, self-reliant, with good willpower and perseverance.” Third, his professional role as a physical education teacher also imposed a sense of self-discipline, as he mentioned, “My job as a sports teacher is another important factor driving my participation.”

### 4.2. Intermediate Period (2003–2010): Reflection on Health and Transformation

Table 2 presents the results of the psychological characteristics analysis of Mr. A during the intermediate period.

As shown in Table 2, the results indicate that during the intermediate period, Mr. A’s physical capacity underwent changes due to aging, prompting a redefinition of the meaning of running. His psychological characteristics shifted from competition-driven to health-focused transformation. This shift is manifested in two main aspects.

Firstly, Mr.A prioritized health and sustainable running. He increasingly recognized the importance of health, moving from a competition-oriented to a broader, public-oriented perspective. Instead of viewing running solely as a competitive challenge, he embraced it as a sustainable health practice. He mentioned that as he grew older, he stopped pushing himself to the limit. Instead, his focus shifted to maintaining his health and performing to the best of his ability given his age. He emphasized that health now comes first, and he prioritizes body maintenance. This reflects his growing emphasis on adaptability and long-term health over competitive outcomes. Additionally, Mr. A began to focus on the scientific and personalized nature of training, highlighting the importance of adjusting training plans based on his physical condition and goals. He mentioned, “I developed scientific training methods to avoid injuries.”

Secondly, his professional role as a physical education teacher activated his extrinsic motivation, leading to a shift from “running fast” to “teaching well.” Mr. A underscored the demonstrative role of a sports teacher in running and began focusing on the educational and social value of running. He stated, “It’s embarrassing for a sports teacher to get injured while running.” He advocated for “joyful running, happy running, and gradual progress, emphasizing the importance of building a solid foundation before pursuing better performance.” His professional responsibility as a sports teacher drove him to focus on scientific training and healthy running, setting a positive example for his students and fellow runners.

### 4.3. Master(s) Level Athlete Period (2010–Present): Perpetuating the Marathon Legacy and Lifelong Running

Table 3 displays the results of the psychological characteristics analysis of Mr. A during the master(s) level athlete period.

As shown in Table 3, the results indicate that during the master(s) level athlete period, Mr. A’s psychological characteristics shifted from personal achievement to perpetuating the marathon legacy and lifelong running. This transition aligns with self-determination theory ([8]), which suggests that long-term engagement in an activity is driven by intrinsic motivation and the pursuit of self-actualization and meaningfulness. Mr. A’s core psychological features during this stage were characterized by perpetuating the marathon legacy and lifelong running.

During this period, Mr. A’s psychological characteristics gradually oriented toward perpetuating the marathon legacy, viewing running as a form of cultural transmission and social responsibility. This shift was driven by three main factors:(1)Age and Goal Transition. Mr. A redirected his running goals from performance achievement to health and legacy. He stated, “At 56, I no longer focus on athlete performance outcomes,” indicating his acceptance of age-related physical changes and a shift toward long-term health and inspiration for the next generation;(2)Educator and Promoter’s Sense of Mission. Mr. A emphasized his role as an educator, noting, “As a teacher, I must set an example for my students and fellow runners.” He leveraged the “appeal” of running to help others develop a passion for the sport, using public welfare training camps and marathon courses to disseminate running culture more broadly;(3)Philanthropy and Social Contribution. Mr. A actively engaged in public welfare activities, viewing running as a means of social contribution. He stated, “Running has a unique charm that helps many people fall in love with it and build deep friendships.” He also facilitated resource connections, such as donations from Xtep, to support the next generation, demonstrating his philanthropic spirit and social responsibility. He mentioned, “I conduct public welfare training camps at the sports center every Monday and Friday evening, previously on Saturdays and Sundays at the Xiamen University track,” highlighting activities that not only benefited others but also enhanced his social standing as a marathon influencer.

On the other hand, Mr. A regarded running as a lifelong endeavor, emphasizing the power of persistence and habit. He set a lifelong goal of running until the age of 99, symbolized by his permanent number 599 at the Xiamen Marathon. This goal not only motivated him personally but also became a part of his social influence. He noted, “Make habits second nature”, indicating that running had become an integral part of his daily life. He also expressed a positive attitude toward growth, saying, “It’s about embracing opportunities and striving for continuous improvement.” Additionally, his professional spirit as a physical education teacher was a significant factor in promoting lifelong running. He stated, “As a sports teacher, I practice what I preach and have been dedicated to my profession for 35 years”, highlighting that his professional identity served as both a driving force and a critical support for his long-term commitment to running.

## 5. Discussion

Marathon running is a highly demanding sport that has attracted substantial attention in sports psychology due to its unique requirements on participants’ psychological processes. However, existing research predominantly focuses on singular motivational theories, such as achievement motivation or self-determination theory, which are insufficient in capturing the complex psychological dynamics involved in long-term marathon participation. Moreover, the majority of studies employ cross-sectional designs, lacking longitudinal tracking of individual psychological changes over time. This limitation constrains our understanding of the dynamic interplay and evolution of motivations.

To address these limitations, this study longitudinally examines the psychological evolution of a master(s) level marathon runner and proposes a novel “motivation-physicality-society” interaction model. The model reveals the dynamic changes in motivation from the novice to the master(s) level athlete stage and emphasizes the interplay between internal and external factors. Additionally, the model explores the dynamic reconstruction of identity, where experienced runners shift from individual achievement to social legacy. Finally, this study integrates these findings into a comprehensive framework. The “motivation-physicality-society” interaction model fills this gap by integrating individual psychological evolution with broader sociocultural transformations. Through a longitudinal analysis of an elite runner’s career spanning over 20 years, we demonstrate how motivations shift from external competition to intrinsic legacy-building, how bodily constraints catalyze identity reconfiguration, and how individual practices reciprocally shape and are shaped by sociocultural structures. This model challenges static psychological frameworks and provides a holistic perspective for understanding long-term sports participation. Below, we synthesize these insights through three interrelated dimensions, supported by expanded theoretical linkages and empirical evidence.

### 5.1. Dynamic Interplay of Motivation: From Singular Motives to Balanced Integration

Consistent with previous research, Mr. A’s psychological changes conform to the views of achievement motivation ([17]) and self-determination theory ([8]). Furthermore, this study employs a three-stage model to elucidate the dynamic interplay of motivation. During the novice stage, external achievement motivation (e.g., competitive performance and social recognition) prevails, aligning with traditional explanations of achievement motivation in competitive sports. However, the emergence of “health reflection” in the intermediate stage and the “legacy mission” in the advanced stage highlight the progressive strengthening of intrinsic motivation (e.g., self-actualization and altruism). These findings resonate with [27]’s ([27]) hierarchical model of motivation, which posits that motivation can be internalized from situational (e.g., external competitive goals in the novice stage) to integrated (e.g., altruistic missions in the advanced stage). This discovery challenges the entrenched dichotomy of “internal versus external” in motivation research, suggesting that long-term participation in marathon running is driven by a dynamic equilibrium between internal and external factors.

Additionally, [16] ([16]) demonstrated, through their longitudinal study of elite athletes, that the dynamism of motivation is significantly influenced by life cycle events, such as career transitions. This finding is consistent with the motivational shift observed in the intermediate stage of our study, where the identity of an educator triggered changes in motivation. Furthermore, the decline in physical function (e.g., aging) and changes in social roles (e.g., the role of an educator) serve as triggering mechanisms for motivational transformation, providing empirical support for the integration of biological and sociological perspectives.

According to the Selective Optimization with Compensation (SOC) theory proposed by [1] ([1]), successful aging is an adaptive process involving selection, optimization, and compensation. Specifically, individuals facing age-related gains and losses achieve optimal adaptation by selectively allocating resources, optimizing opportunities for goal attainment, and compensating when necessary. This theoretical framework emphasizes the proactivity and adaptability of individuals during the aging process and how they strategically adjust to cope with the challenges of aging. For instance, research by [18] ([18]) demonstrated that the motivations of older marathon runners are more closely related to health and social interaction rather than competitive achievement. This finding aligns with the SOC model, highlighting the adaptive nature of motivation in later life.

Additionally, [10]’s ([10]) theory of psychosocial development stages provides theoretical support for the dynamic understanding of motivation ([10]). This theory posits that individuals encounter specific psychological conflicts at different life stages. For example, in late adulthood, the conflict between “integrity versus despair” is central. The successful resolution of these conflicts leads to psychological fulfillment, while failure may result in regret and despair. This dynamic understanding of the life cycle aligns closely with the motivational transitions observed in this study, further supporting the dynamic and context-dependent nature of motivation.

In summary, this study not only reveals the dynamic interplay of motivation but also integrates multiple theoretical frameworks to provide a more comprehensive perspective on the motivations underlying long-term participation in sports.

### 5.2. Dynamic Reconstruction of Identity: From Individual Achievement to Social Legacy

This study reveals that experienced runners’ psychological characteristics shift from “personal breakthrough” to “social legacy,” with their identity evolving from “athlete” to “educator” and finally to “cultural promoter.” This process engages in dialogue with [25]’s ([25]) social identity theory, which emphasizes that individual behavior is largely influenced by perceived intergroup relationships and that social identity is a dynamic construct embedded in social structures ([25]).

Moreover, this finding resonates with [23]’s ([23]) identity control theory, which posits that identity is not a static social affiliation but a dynamic, continuously adjusted process. This dynamism enables individuals to switch between different social roles and adjust their behaviors based on social feedback. For example, [9] ([9]) demonstrated that the identity shift from “conqueror” to “environmental advocate” among mountaineers requires social interaction as a catalyst. This aligns closely with the process observed in this study, where runners reinforce their identity as “cultural promoters” through philanthropic networks.

In summary, this study not only uncovers the dynamic interplay of motivation but also integrates multiple theoretical frameworks to provide a comprehensive understanding of the motivations underlying long-term sports participation.

In addition to the findings discussed earlier, [5]’s ([5]) Optimal Distinctiveness Theory can explain how runners balance their unique identities (e.g., the symbolic significance of lifelong bib numbers like No. 599) with integration into the broader running culture. This mechanism transcends the static descriptions of group belongingness found in prior research and reveals the dynamic creativity of social identity in long-term sports participation.

Moreover, the case study of “lifelong running” goals (e.g., “running until age 99”) integrates individual life trajectories with the cultural symbols of marathon running (e.g., permanent bib numbers). This integration provides new pathways for exploring the intersection of sports participation and life meaning. This “meaning production” mechanism not only enriches the understanding of identity but also emphasizes the productive nature of identity rather than its attributional nature. Runners actively construct a new identity as “marathon culture promoters” through practices such as philanthropic training and resource coordination, thereby reframing the meaning of running from personal challenge to social contribution.

### 5.3. Tripartite Interaction Model: Embedding Individual Evolution in the Sociocultural Context

Based on the above discussion, a three-dimensional interaction model of “motivation-physicality-society” can be developed, revealing the dynamic relationship between individual psychological evolution in marathon running and socio-cultural transformation. This is specifically manifested in the following three aspects.

Firstly, physicality as a mediator. In the psychological evolution of marathon runners, physicality functions both as a vessel for physiological capabilities and as a medium for conveying cultural significance. During the intermediate stage, physiological decline—characterized by aging and increased susceptibility to injury—necessitates a prioritization of health. For instance, Mr. A adopted scientific training methods to mitigate injury risk, aligning with the Selective Optimization with Compensation (SOC) theory proposed by [1] ([1]). This theory posits that individuals optimize their goals and employ compensatory strategies to maintain engagement in the face of physiological limitations. Mr. A’s aspiration, “I want to run to 99,” exemplifies the transition from biological constraints to cultural aspirations, reflecting a shift from managing aging-related limitations to embracing lifelong running as a cultural pursuit.

Furthermore, Mr. A’s bodily practices extend beyond physiological adaptation to encompass cultural significance. His involvement in public events, such as filming promotional videos for the 2024 Paris Olympics, exemplifies the “physicality as project” concept introduced by [7] ([7]). This concept frames the physicality as both a physiological entity and a cultural production tool. Mr. A’s practices not only compensate for physiological limitations but also convey sociocultural meanings through symbolic value. As noted by [7] ([7]), the physicality is both shaped by social interactions and serves as a carrier of cultural significance. Mr. A’s promotion of healthy living through running and his efforts to enhance public awareness of marathon culture exemplify how bodily practices can embody and propagate cultural values.

Secondly, society as a catalyst. Social factors play a pivotal role in the psychological evolution of marathon runners. Occupational roles and social capital provide critical resources for identity reconstruction. For instance, Mr. A’s role as a physical education instructor prompted a shift from “running fast” to “teaching well,” consistent with [29]’s ([29]) findings that career transitions trigger identity reconfiguration. Additionally, Mr. A’s social networks, including student groups and corporate partners, provided social capital that facilitated his transition from athlete to social influencer. [15]’s ([15]) social capital theory posits that social networks, by offering resources and support, help individuals achieve identity transformation. Mr. A’s public welfare training camps and brand collaborations exemplify this social capital, supporting his running career and expanding his social influence.

Moreover, the sociocultural context profoundly impacts the psychological evolution of marathon runners. The transition of China’s marathon culture from elitism to mass participation mirrors Mr. A’s personal journey, reflecting [12]’s ([12]) structuration theory, which emphasizes the bidirectional interaction between individual practices and social structures. Mr. A’s public welfare training camps both responded to and propelled the trend of mass participation in marathons in China. This bidirectional interaction not only altered the perception of marathon culture but also reshaped the identity of runners. For example, marathon runners develop a unique social identity and sense of belonging through social interactions. Mr. A’s integration of personal identity with sociocultural transformation through public welfare training camps and brand collaborations exemplifies the dynamism of such social interactions. It should be emphasized that before Mr. A started running, these training camps did not exist and were all founded by Mr. A. This must be a motivating factor that motivates Mr. A to work hard to promote the marathon and position it as a lifelong goal.

Thirdly, motivation as a dynamic force. Motivation plays a central role in the psychological evolution of marathon runners. Shifting from external competitive motives (e.g., pursuing performance and social recognition) in the novice stage to intrinsic legacy-building motives (e.g., social contribution and lifelong running) in the master(s) level athlete stage, motivation both drives and is influenced by changes in the body and society. For example, Mr. A’s focus on competitive achievements in the novice stage aligns with [17]’s ([17]) achievement motivation theory, which posits that early participation is primarily driven by external achievement motives. However, as he aged and experienced physical decline, his motivation gradually shifted toward health and sustainable running, consistent with [18]’s ([18]) findings that older runners prioritize health over competition ([18]).

In the master(s) level athlete stage, Mr. A’s motivation further evolved toward intrinsic legacy-building. His philanthropic endeavors, such as organizing public welfare training camps and securing donations for young runners, reflect [27]’s ([27]) hierarchical model of motivation, which suggests that motivation can be internalized from external goals to intrinsic values. Additionally, his permanent bib number (No. 599) symbolizes [10]’s ([10]) stage of “generativity versus stagnation,” emphasizing legacy as a psychological imperative. This intrinsic motivation not only sustained his running habit but also propelled sociocultural transformation. For example, [8] ([8]) noted that when intrinsic motivation aligns with societal values, individual engagement becomes more enduring and meaningful.

Subsequently, with regard to cultural transformation and the interplay between the individual and society, this model is embedded in the context of China’s marathon culture, transitioning from elitism to mass participation. The psychological evolution of individual runners (e.g., from competition to legacy) resonates with the societal shift in the perception of marathon running (e.g., from a competitive sport to a healthy lifestyle). This interplay between individual and society corroborates the core proposition of [12]’s ([12]) structuration theory, which emphasizes the dynamic duality between social structure and individual practice. This provides a classic case for the study of the interplay between individual practice and social structure in sports sociology.

In summary, this study proposes a three-dimensional interaction model of “motivation-physicality-society” to reveal the dynamic evolution of marathon runners at psychological, physical, and social levels and emphasizes the interplay between individual practice and social structure.

## 6. Conclusions

This study provides a comprehensive longitudinal analysis of the psychological evolution of a master(s) level athlete, revealing the dynamic interplay between motivation, identity, and sociocultural context across different life stages. The findings challenge static theories of sports psychology by proposing a novel “motivation-physicality-society” interaction model. This model highlights how long-term marathon participation is influenced by the interplay of physiological changes, social roles, and cultural shifts. The study demonstrates that elite runners transition from a competitive mindset driven by external achievement motives to a legacy-building focus rooted in intrinsic values and social contributions. This research underscores the importance of understanding the dynamic nature of motivation and identity in long-term sports participation and provides a theoretical foundation for optimizing psychological support systems in marathon running.

## 7. Strengths

(1)(This study provides a unique longitudinal perspective by examining the psychological evolution of a marathon runner across three distinct stages—novice, intermediate, and master(s) level athlete. This approach allows for a nuanced understanding of the dynamic changes in motivation, identity, and social roles over time, which is a significant improvement over cross-sectional studies that capture only a single point in time;(2)The study proposes a novel “motivation-physicality-society” interaction model. This model integrates frameworks from psychology (e.g., SOC theory), physiology (e.g., sociology of the body), and sociology (e.g., structuration theory). It addresses the fragmented explanations in sports psychology. For instance, it combines [1]’s ([1]) focus on aging with [25]’s ([25]) emphasis on social identity. It demonstrates how physiological decline and social roles jointly drive motivational changes;(3)This study situates its findings within the broader context of China’s marathon culture, highlighting how individual experiences are shaped by and contribute to socio-cultural transformations. This contextualization enriches the understanding of marathon running as a socio-cultural phenomenon and provides valuable insights for similar studies in other cultural settings.

## 8. Limitations

(1)Single-Case Study Design. The reliance on a single case limits the generalizability of the findings. While the subject’s extensive experience and influence make him an ideal candidate for in-depth analysis, the results may not be representative of all marathon runners, particularly those with different backgrounds or experiences. Future research should consider multi-case studies or larger samples to enhance generalizability;(2)Cultural Specificity. The study’s focus on the Chinese cultural context may limit the applicability of the findings to other cultural settings. Future research should explore whether similar psychological trajectories and mechanisms are observed in marathon runners from different cultural backgrounds, thereby enhancing the generalizability of the “motivation-physicality-society” interaction model;(3)The three-stage model of exploring the dynamic interaction of motivation is useful, but it seems that there are other factors at play, especially during the “veteran/master level” period, which can be further explored in future research.

## 9. Future Research Directions

Future studies should address these limitations by adopting multi-case designs and conducting cross-cultural comparisons. Longitudinal cohort studies could provide further insights into the generalizability of the findings and the broader implications for long-term engagement in marathon running. Such research would contribute to a more comprehensive understanding of the psychological dynamics underlying sustained participation in sports and inform the development of targeted interventions to support runners’ well-being and performance.

## Figures and Tables

**Table 1 behavsci-15-00893-t001:** The results of the psychological characteristics analysis of Mr. A during the novice period.

Selective Coding	Axial Coding	Open Coding/Cases
Competitive mindset	Self-demands as a professional athlete	My first marathon was driven by my lifelong passion for running, my background as a professional runner, and my academic focus on running. Despite being a specialist in the 800m track event and being 35 years old, I felt compelled to participate actively in the inaugural Xiamen Marathon.
	Tenacious spirit	Special promotional video for the 2024 Paris Olympics on CCTV (Ode to the Chinese Sports Spirit), filmed by me. The lifelong spirit of hard work serves as the foundation for continuous marathon running.
		In middle school, I was so determined to win that I would have pushed myself to the limit.
		Running a marathon is not just about running; it’s about cultivating perseverance and a strong will.
Self -breakthrough	Pride in completing arduous tasks	Each full marathon is tough, but finishing it brings immense satisfaction and a sense of accomplishment.
	Adhered to a self-reliant	I may encounter minor injuries and sometimes get sick, but my name is hardworking, self-reliant, with good willpower and perseverance.
	Professional self-restraint	My job as a sports teacher is another important factor driving my participation.

**Table 2 behavsci-15-00893-t002:** The results of the psychological characteristics analysis of Mr. A during the intermediate period.

Selective Coding	Axial Coding	Open Coding/Cases
Health-focused transformation	Health priority and sustainable running	As I got older, I no longer pushed myself to the limit. My focus shifted to maintaining health and performing to the best of my age.
		Now, health comes first. I prioritize body maintenance.
		I developed scientific training methods to avoid injuries.
	Professional responsibility and happy running	It’s embarrassing for a sports teacher to get injured while running.
		joyful running, happy running, and gradual progress, emphasizing the importance of building a solid foundation before pursuing better performance.

**Table 3 behavsci-15-00893-t003:** The results of the psychological characteristics analysis of Mr. A during the master(s) level athlete period.

Selective Coding	Axial Coding	Open Coding/Cases
Perpetuating the Marathon Legacy	Age and Goal Transition	At 56, I no longer focus on athlete performance outcomes
	Educator and Promoter’s Sense of Mission	As a teacher, I must set an example for my students and fellow runners
	Philanthropy and Social Contribution	Running has a unique charm that helps many people fall in love with it and build deep friendships.
		I conduct public welfare training camps at the sports center every Monday and Friday evening, previously on Saturdays and Sundays at the Xiamen University track.
		I have connected Wang Rui, Deputy Secretary General of the Student China Care for the Next Generation Committee, to Wang Qiang, Special Assistant to the President of Xtep, and organized Xtep Group to donate clothes and backpacks to the next generation.
		I have gained many friendships and helped many people fall in love with running. A philanthropic heart. Work harder, there is a place for success, and become a marathon influencer.
Lifelong running	Running as a lifelong endeavor	I want to run to 99.
		Running a marathon, unconsciously persisting for 10 years, sharpening a sword in 10 years, and choosing the 599 permanent number.
	Persistence and habit	Make habits second nature. And it’s also fate, it’s good that it’s here, and with more effort, it will become better and better.
	Leadership	As a sports teacher, I practice what I preach and have been dedicated to my profession for 35 years
		As a teacher, I need to set an example and lead by example when facing college students in physical education courses and fellow runners in public marathon training camps.

## Data Availability

The data of the present study are available from the corresponding author upon reasonable request.

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
