# Peer review of "From Novice to Master(s) Level Athlete: A Longitudinal Analysis of Psychological Changes in a Marathon Runner Completing 119 Marathons"

_behavsci, 2025, doi:10.3390/bs15070893_

Round 1
Reviewer 1 Report
Comments and Suggestions for Authors
Dear authors and editors,
The article deals with a very relevant topic and uses an innovative and interesting longitudinal research method. A sufficient theoretical overview has been carried out and the results and conclusions have been clearly presented.
However, I would like to make a few comments and suggestions:
a) the article does not pose a hypothesis, perhaps it would be interesting to know if the researchers had any hypothetical insights that could be presented in the article;
b) in the Participants section, the periods when the interviews were conducted should be more specifically indicated (e.g. every year at the same time, or before, after the competition, etc.). I would think that the text in lines 137-143 in this section is redundant (it should be indicated not here, but in the Procedure section). It would also be interesting if more information about the subject was indicated in this section, e.g. changes in physical health, etc.
c) at least a few examples of interview questions and how long each interview lasted should be provided in the Procedure section.
d) Interesting and innovative the title of a "motivation-body-society" interaction model (although the model itself is very relevant and well-founded), e.g. according to the data presented, perhaps the "body" part of the model
could be called "health" or "physicality"- but this is just a suggestion.
I think these comments are not essential and can be easily corrected.
Author Response
|
Comments 1: the article does not pose a hypothesis, perhaps it would be interesting to know if the researchers had any hypothetical insights that could be presented in the article |
||
|
Response 1: Thank you for pointing this out. We agree with this comment. Therefore, we have added the following content,”In light of the preceding discussion, we posit the following hypothesis: Hypothesis 1: The interplay among motivation, the physical self, and societal factors offers a more comprehensive and dynamic explanation of the psychological evolution of marathon runners. This framework elucidates the reciprocal relationships between motivation, physical changes, and social influences, as well as their collective impact on runners' psychological states.”(Line 107-113) Based on the literature review above, the marathon runner who completed 119 marathons experienced psychological changes from the novice stage through the intermediate stage to the master level. The following hypotheses are proposed: Hypothesis 2: There are differences in the psychological characteristics of marathon runners across different stages from novice to master level. Hypothesis 3: Novice marathon runners are primarily driven by external achievement motivation, such as the pursuit of competitive performance. Hypothesis 4: Intermediate-level marathon runners shift their motivation towards prioritizing health and reconstructing their identity. Hypothesis 5: At the master level, marathon runners are mainly motivated by the mission of perpetuating the marathon spirit and lifelong running.(Line133-143) |
||
|
Comments 2: in the Participants section, the periods when the interviews were conducted should be more specifically indicated (e.g. every year at the same time, or before, after the competition, etc.). I would think that the text in lines 137-143 in this section is redundant (it should be indicated not here, but in the Procedure section). It would also be interesting if more information about the subject was indicated in this section, e.g. changes in physical health, etc. |
||
|
Response 2: Agree. The text in lines 137-143 in this section has already been moved to the Procedure section.(Line 178-185) |
||
|
Comments 3: at least a few examples of interview questions and how long each interview lasted should be provided in the Procedure section. |
||
|
Response 3: Agree. Therefore, we have added the following content, “interviews lasted approximately one hour. The interview questions primarily included: "What were your initial motivations for running marathons?" "How have these motivations changed over time? Can you delineate these changes by stages?" "What has sustained your commitment to complete 119 marathons?"(Line168-171) |
||
|
Comments 4: Interesting and innovative the title of a "motivation-body-society" interaction model (although the model itself is very relevant and well-founded), e.g. according to the data presented, perhaps the "body" part of the model I think these comments are not essential and can be easily corrected. |
||
|
Response 4: We accept this comment and have changed “body” to “physicality”(Line22, 358, 363, 455, 458, 459, 471, 472, 475, 535, 543, 558, 581) |
||
|
4. Response to Comments on the Quality of English Language |
||
|
Point 1: The English is fine and does not require any improvement. |
||
|
Response 1: Thank you very much. |
||
|
Reviewer 2 Report
Comments and Suggestions for Authors
From Novice to Veteran: A Longitudinal Analysis of Psychological Changes in a Marathon Runner Completing 119 Marathons
Thank you for opportunity to review this paper. As a runner I took great interest in this paper and enjoyed reading it immensely. I offer the below comments in good faith – please do not be offended. Most of my comments relate to a lack of clarity and detail in some instances. Finally, I hope Mr A fulfils his ambition to run until he is 99.
Line 32: Suggest changing that the sport is ‘psychological test’ to ‘psychological challenge’.
Line 50-55: I believe more clarity is needed here. For instance, you state ‘runners’, which assumes that there are many. Yet the study focuses on one runner only. You could alter this to say that the review explored the number of participants (years of competing, etc), which then sets the scene as to why only one runner was reviewed.
Line 63: Space needed before citation [9].
Line 85-86: This is a little unclear. You state that ‘despite existing insights into the psychological characteristics of marathon runners, most studies have focused on specific stages or groups, lacking longitudinal tracking of psychological changes.’. Is this for psychological changes during one marathon or throughout many?
Line 110: Can some examples be provided in what specific cultural contexts exist?
Line 122: All that is stated here is informative and quite correct, but it is important to acknowledge that not everyone pursues marathon running long-term – there is a significant drop-off at varying timepoints, particularly those who complete a marathon as ‘bucket list’ achievement. I would suggest that this is acknowledged.
Line 138: Suggest changing ‘veteran’ to ‘master(s) level athlete.
Line 145: I believe you need to be very clear about your association with Mr. A given that you are colleagues. You run the risk of potential bias if the relationship is not fully explained. I’m assuming that there is ethical clearance for this study?
Line 159: I don’t believe that you can say that you observed Mr A’s marathon participation given how long he has been doing them (unless you attended everyone of them). I would clarify this statement.
Line 173: How much experience did the five experts have with coding? What was the process of synthesising the various responses and how were you involved in this process? Can this process be fully explained as it’s unclear how final themes are actually developed. Saying ‘collective discussion’ is vague and doesn’t define how the codes were formed.
Line 196: Was Mr A coached during his middle school years? It would be interesting to know if his initial motivation was intrinsic or extrinsic.
Line 231-232: Can other quotations be used? As it stands quotations are duplicated in the text and the tables. It doesn’t really justify having both.
Line 242: I find this a very interesting quotation, “It's embarrassing for a sports teacher to get injured while running.’ Is more information available? Why is it embarrassing? Running takes a tremendous toll on the body and sometimes injuries cannot be helped. Does this Mr A run whilst injured?
Table 3. As a fellow runner, I appreciate that Mr A ‘wants to run until 99’. I wonder what is the driving factor to do this. Is there a connection to nature or a freedom associated with this desire?
Line 263: Performance is a subject measure. Was this considered? For instance, ‘At 56, I no longer focus on performance outcomes.’. But health and vitality could be a performance outcome. Performance is not necessarily competitive based.
Line 267: Reading this again, ‘Leadership’ may be a better classification in Table 3 rather than professional spirit. It would appear that Mr A’s desire to set a standard for his students is a leadership trait.
Line 292: I refer to my earlier comment in that marathon running isn’t necessarily a ‘long-term …’ sport for some.
Line 318-319: Judging by Mr A’s responses, I would suggest that achievement motivation [20] and self-determination theory are relevant in your study. Both can be linked to responses by Mr A.
Line 320: The three-stage model to explore ‘the dynamic interplay of motivation’ is useful, but there appears to be other factors at play here, particularly during the ‘veteran/master’s’ years.
Line 440: ‘… integration of personal identity with sociocultural transformation through public welfare training camps and brand collaborations exemplifies the dynamism of such social interactions.’ Were such training camps in existence when Mr A commenced marathon run? Has the creation of these camps acted as a source of motivation?
Line 470-471. Full stop missing after ‘practice’.
Author Response
For research article
|
Response to Reviewer 2 Comments |
||
|
1. Summary |
|
|
|
Thank you very much for taking the time to review this manuscript. Please find the detailed responses below and the corresponding revisions/corrections highlighted/in track changes in the re-submitted files. |
||
|
2. Questions for General Evaluation |
Reviewer’s Evaluation |
Response and Revisions |
|
Does the introduction provide sufficient background and include all relevant references? |
Yes/Can be improved/Must be improved/Not applicable |
Yes, the introduction provide sufficient background and include all relevant references |
|
Are all the cited references relevant to the research? |
Yes/Can be improved/Must be improved/Not applicable |
Yes, all the cited references relevant to the research |
|
Is the research design appropriate? |
Yes/Can be improved/Must be improved/Not applicable |
Yes, the research design appropriate |
|
Are the methods adequately described? |
Yes/Can be improved/Must be improved/Not applicable |
Yes, the methods adequately described |
|
Are the results clearly presented? |
Yes/Can be improved/Must be improved/Not applicable |
Yes, the results clearly presented |
|
Are the conclusions supported by the results? |
Yes/Can be improved/Must be improved/Not applicable |
Yes, the conclusions supported by the results |
|
3. Point-by-point response to Comments and Suggestions for Authors |
||
|
Comments 1: Line 32: Suggest changing that the sport is ‘psychological test’ to ‘psychological challenge’. |
||
|
Response 1: We greatly appreciate and accept this valuable suggestion from the expert. We have changed the sentence,“This sport is not only a physical challenge but also a psychological test.” to “This sport is not only a physical challenge but also a psychological test.”(Line 33) |
||
|
Comments 2: Line 50-55: I believe more clarity is needed here. For instance, you state ‘runners’, which assumes that there are many. Yet the study focuses on one runner only. You could alter this to say that the review explored the number of participants (years of competing, etc), which then sets the scene as to why only one runner was reviewed. |
||
|
Response 2: Agree. We have changed ”in marathon runners” to “ in the number of participants ”(Line 53-54) |
||
|
Comments 3: Space needed before citation [9]. |
||
|
Response 3: We greatly appreciate and accept this valuable suggestion from the expert. We have added the space before citation [9] (Line 65) |
||
|
Comments 4: Line 85-86: This is a little unclear. You state that ‘despite existing insights into the psychological characteristics of marathon runners, most studies have focused on specific stages or groups, lacking longitudinal tracking of psychological changes.’. Is this for psychological changes during one marathon or throughout many? |
||
|
Response 4: The psychological changes here are from multiple marathons. We have made modifications to this, as follows::“Despite existing insights into the psychological characteristics of marathon runners, most studies have focused on specific stages or groups, lacking longitudinal tracking of psychological changes in runners who participate in multiple marathons over an extended period. ”(Line 87-88) |
||
|
Comments 5:Line 110: Can some examples be provided in what specific cultural contexts exist? |
||
|
Response 5: We greatly appreciate and accept this valuable suggestion from the expert. The manuscript does not emphasize the specificity of culture in the psychological changes of marathon runners. Therefore, we have deleted the following sentence:“Second, research has predominantly focused on specific cultural contexts, with a lack of cross-cultural comparisons”. |
||
|
Comments 6:Line 122: All that is stated here is informative and quite correct, but it is important to acknowledge that not everyone pursues marathon running long-term – there is a significant drop-off at varying timepoints, particularly those who complete a marathon as ‘bucket list’ achievement. I would suggest that this is acknowledged. |
||
|
Response 6: We appreciate your reminder, it makes perfect sense, and we acknowledge it. In our manuscript, the case we selected is the psychological changes of people who have been running marathons for a long time or even their entire lives, targeting only one group. We can delve deeper into the churn situation you mentioned in future research. |
||
|
Comments 7:Line 138: Suggest changing ‘veteran’ to ‘master(s) level athlete. |
||
|
Response 7: Agree. We have changed all instances of “veteran” to “master(s) level athlete” in the text.(Line 2, 12, 46, 231, 300, 303, 304, 306, 359, 508, 515.540, 554 ) |
||
|
Comments 8:Line 145: I believe you need to be very clear about your association with Mr. A given that you are colleagues. You run the risk of potential bias if the relationship is not fully explained. I’m assuming that there is ethical clearance for this study? |
||
|
Response 8: Agree. We add the following contents:“In addition, to address potential biases arising from the first author's professional relationship with interviewee A, we implemented several measures to ensure the objectivity and impartiality of the interview. Firstly, we adopted anonymous recording during the interview process. Additionally, we invited a researcher, who has no direct connection with either the first author or the interviewee, to observe and document the interview. This arrangement helps to maintain the integrity of the interview data. In the data analysis phase, we employed multiple methods to cross - validate the data. For instance, we analyzed and compared publicly available television interviews and newspaper reports about Mr. A to minimize interpretative biases that might stem from personal relationships. Moreover, four experts were invited to collaborate with the first author in the data analysis process, further ensuring the objectivity of the research findings.
It should be noted that this study obtained ethical approval from our institution, which deemed there were no ethical concerns. Mr. A, a veteran teacher nearing retirement, has a profound passion for marathons. He expressed his willingness to share all relevant information openly and honestly, without any concealment or sensitivity issues. These factors collectively alleviate potential reader concerns regarding ethical issues.(Line 203-220) |
||
|
Comments 9: Line 159: I don’t believe that you can say that you observed Mr A’s marathon participation given how long he has been doing them (unless you attended everyone of them). I would clarify this statement. |
||
|
Response 9: Agree. We have deleted this sentence and the paragraph in which it is located.“Moreover, the researchers observed Mr. A's participation in marathon events, teaching courses, and public welfare activities. The observations focused on his performance in competitions, interactions during teaching, and communication with runners and students. Using detailed behavioral coding and descriptive recording methods, the researchers accurately captured his psychological states and behavioral manifestations in different contexts. The combination of observational and interview data enriched and enhanced the empirical support for the study.”(Line 178) |
||
|
Comments 10: Line 173: How much experience did the five experts have with coding? What was the process of synthesising the various responses and how were you involved in this process? Can this process be fully explained as it’s unclear how final themes are actually developed. Saying ‘collective discussion’ is vague and doesn’t define how the codes were formed. |
||
|
Response 10: Agree.We add the following contents:“To ensure the credibility and validity of the findings, the first author convened a five-member analysis team, comprising four additional experts, to undertake the data coding process. Each member of this team is a doctoral candidate with a background in conducting interview-based research and a publication record in this domain, and also with extensive experience and a strong theoretical foundation in marathon studies. Their collective expertise ensures a robust and reliable coding process for this research endeavor. ”(Line194-200) 。 In the coding process, open coding was initially conducted to extract sentences from Mr. A's marathon running experiences that were linked to his psychological feelings, such as his statement in this study's interview: "Now, health comes first. I prioritize body maintenance." These sentences were then identified as content for further analysis and comparison. Subsequently, axial coding was performed to refine the extracted sentences from the open coding, resulting in more concise concepts like "Health priority and sustainable running." Finally, selective coding was applied to compare and distill the axial coding outcomes into more abstract meaning units, such as "Health-focused transformation."”(Line 221-229) |
||
|
Comments 11: Line 196: Was Mr A coached during his middle school years? It would be interesting to know if his initial motivation was intrinsic or extrinsic. |
||
|
Response 11: Agree. We will make the following statement in the manuscript: “Mr. A had not received formal coaching for marathon running but had been trained in sprinting. This background in sprint training can be considered as a form of relevant instruction. His initial motivation was a combination of external and internal factors. Externally, his coach's expectations played a role, while internally, he aspired to achieve good athlete performance results.”(Line148-152) to provide readers with a clearer understanding of the research subject's background and to clarify the scope of applicability of the study's findings. |
||
|
Comments 12: Line 231-232: Can other quotations be used? As it stands quotations are duplicated in the text and the tables. It doesn’t really justify having both. |
||
|
Response 12: Agree. We changed the expressions to “He mentioned that as he grew older, he stopped pushing himself to the limit. Instead, his focus shifted to maintaining his health and performing to the best of his ability given his age. He emphasized that health now comes first and he prioritizes body maintenance.” (Line 283-286) |
||
|
Comments 13:Line 242: I find this a very interesting quotation, “It's embarrassing for a sports teacher to get injured while running.’ Is more information available? Why is it embarrassing? Running takes a tremendous toll on the body and sometimes injuries cannot be helped. Does this Mr A run whilst injured? |
||
|
Response 13: We visited Mr. A again and he emphasized: "As a physical education teacher, it's not impossible to get injured. If a physical education teacher frequently gets injured while running, then their running skills and techniques need to be reconsidered by themselves. Because physical education teachers are role models for students and the power of role models. For example, if a doctor frequently gets sick, they may have deficiencies in their own maintenance, and patients will definitely like to seek medical treatment from doctors who frequently get sick. In addition, as a teacher, not an athlete, we need to pay more attention to safety when conducting our own physical exercise. We should try not to take risks when practicing dangerous movements that are prone to injury. |
||
|
Comments 14:Table 3. As a fellow runner, I appreciate that Mr A ‘wants to run until 99’. I wonder what is the driving factor to do this. Is there a connection to nature or a freedom associated with this desire? |
||
|
Response 14: After follow-up, Mr. A was driven by the constant positive feedback from the outside world. But experts' opinions remind us that in the future, we can further explore the psychological development and changes of marathon runners around nature and a sense of freedom. |
||
|
Comments 15:Line 263: Performance is a subject measure. Was this considered? For instance, ‘At 56, I no longer focus on performance outcomes.’. But health and vitality could be a performance outcome. Performance is not necessarily competitive based. |
||
|
Response 15: Agree. Our expression was incorrect. Mr. A expressed athlete performance outcomes,so we changed‘At 56, I no longer focus on performance outcomes.’ to ‘At 56, I no longer focus on athlete performance outcomes.’( Table 3. and Line 317-318) |
||
|
Comments 16:Line 267: Reading this again, ‘Leadership’ may be a better classification in Table 3 rather than professional spirit. It would appear that Mr A’s desire to set a standard for his students is a leadership trait. |
||
|
Response 16: Agree. We will change the “professional spirit” in Table 3 to“Leadership”.(Table 3) |
||
|
Comments 17: Line 292: I refer to my earlier comment in that marathon running isn’t necessarily a ‘long-term …’ sport for some. |
||
|
Response 17: Agree. We have removed the expression 'long-term...'(Line 347) |
||
|
Comments 18:Line 318-319: Judging by Mr A’s responses, I would suggest that achievement motivation [20] and self-determination theory are relevant in your study. Both can be linked to responses by Mr A. |
||
|
Response 18: Agree. We have revised the relevant contents,“Consistent with previous research, Mr. A's psychological changes conform to the views of achievement motivation [20] and self-determination theory [21]. Furthermore, this study employs a three-stage model to elucidate the dynamic interplay of motivation. (Line 374-376) |
||
|
Comments 19:Line 320: The three-stage model to explore ‘the dynamic interplay of motivation’ is useful, but there appears to be other factors at play here, particularly during the ‘veteran/master’s’ years. |
||
|
Response 19: We accept the opinions of experts and will explain this in our research limitations: ”The three-stage model of exploring the dynamic interaction of motivation is useful, but it seems that there are other factors at play, especially during the "veteran/master level" period, which can be further explored in future research” (Line 582-584) |
||
|
Comments 20:Line 440: ‘… integration of personal identity with sociocultural transformation through public welfare training camps and brand collaborations exemplifies the dynamism of such social interactions.’ Were such training camps in existence when Mr A commenced marathon run? Has the creation of these camps acted as a source of motivation? |
||
|
Response 20: Before Mr. A started running, these training camps did not exist and were all founded by Mr. A. This must be a motivating factor that motivates Mr. A to work hard to promote the marathon and position it as a lifelong goal. We will explain this situation in the manuscript:“It should be emphasized that before Mr. A started running, these training camps did not exist and were all founded by Mr. A. This must be a motivating factor that motivates Mr. A to work hard to promote the marathon and position it as a lifelong goal.”(Line 500-503) |
||
|
Comments 21:Line 470-471. Full stop missing after ‘practice’. |
||
|
Response 21: Agree. We made the modifications as required. (Line 532) |
||
|
4. Response to Comments on the Quality of English Language |
||
|
Point 1: The English is fine and does not require any improvement. |
||
|
Response 1: Thank you very much. |
||
|
5. Additional clarifications |
||
|
We tried our best to improve the manuscript and made some changes marked in red in revised paper which will not influence the content and framework of the paper. We appreciate for editors and reviewers’ warm work earnestly, and hope the correction will meet with approval. Once again, thank you very much for your comments and suggestions. |
||